# Advances in DNA- and RNA-Based Oncolytic Viral Therapeutics and Immunotherapies

**Peter Anto Johnson** [1,2,*], **Alyssa Wu** [1,3,4], **John Christy Johnson** [1,2], **Zachary Schauer** [1], **Terrence Wu** [1], **Francis Fernandes** [1,4], **Reinette Schabert** [1] and **Austin Mardon** [1,2]

1 Antarctic Institute of Canada, Edmonton, AB T5B 0E1, Canada; awu2465@gmail.com (A.W.); jcj2@ualberta.ca (J.C.J.); schauerz@mymacewan.ca (Z.S.); twu342@uwo.ca (T.W.); a46fernandes@uwaterloo.ca (F.F.); eksteenr@mymacewan.ca (R.S.); aamardon@yahoo.ca (A.M.)
2 Faculty of Medicine & Dentistry, University of Alberta, Edmonton, AB T6G 2R3, Canada
3 School of Interdisciplinary Science, McMaster University, Hamilton, ON L8S 4L8, Canada
4 Department of Health Sciences, University of Waterloo, Waterloo, ON N2L 3G1, Canada
* Correspondence: paj1@ualberta.ca

**Abstract:** The role of viruses has been studied extensively for use as curative cancer therapies. However, the natural immunogenicity of viruses and their interaction with the host's immune system needs to be examined to determine the full effectiveness of the viral treatment. The prevalence of cancer is increasing globally, and treatments are needed to support the increasing body of patient care. Oncolytic viral therapies used existing pathogenic viruses, which are genetically modified to not cause disease in humans when administered using a vaccine viral vector. Immunotherapies are another avenue of recent interest that has gained much traction. This review will discuss oncolytic viral approaches using three DNA-based viruses, including herpes simplex virus (HSV), vaccinia virus, and adenovirus; as well as four RNA-based viruses, including reovirus, Newcastle disease virus (NDV), poliovirus, and measles virus (MV). It also examines the novel field of cancer-based immunotherapies.

**Keywords:** viral therapeutics; DNA/RNA-based virus; herpes simplex; vaccinia; adenovirus; poliovirus; reovirus; Newcastle disease virus; measles





## 1. Introduction

Cancer is a broad term used to describe the rapid and uncontrollable growth of abnormal cells that can target any part of the body. Metastasis is an extension of cancer progression, which occurs when the tumorigenic cells spread from the primary organ to external areas in the body [1]. Cancer is one of the largest health burdens worldwide, affecting over 17 million people as of 2018 [2] and causing nearly 10 million deaths in 2020 [1].

Cancer therapies are constantly in development due to the high precedence and increasing demand for treatment to mitigate the symptoms associated with potentially fatal prognoses. The objectives in developing cancer therapeutics are to stop the formation of cancerous cells from growing and spreading, as well as to help control symptoms of advanced cancer progression. Chemotherapy has been a long-used treatment for solid metastatic tumors, despite the cytotoxic risks. Other immunotherapy options for cancer treatment are becoming available providing patients with a safer and effective alternative [3].

Similar to infectious disease vaccines, eliciting an immune response against a cancerous tumor requires the appropriate antigen to be presented to a naive T cell via specialized cells called antigen-presenting cells (APCs). APCs regulate immunity by taking up antigens from the vaccine, showing them to bring about differentiation of naive T cells into memory and effector T cells [4]. Some difficulty arises around using this bodily procedure to the advantage of the vaccine, specifically which antigen is selected. The best properties for

the chosen antigen to have in an ideal situation are cancer cell-specific expression, high immunogenicity, and, ideally, a cancer cell-specific functional dependency [5]. Two possible avenues of development have been proposed, namely the use of tumor-associated antigens and tumor-specific antigens.

Tumor-associated antigens are strongly expressed in cancer cells and make a good target for T cells. However, they are also retained weakly in normal tissues. With this comes a possibility of inducing autoimmune toxicity in normal tissues, such as colitis, hepatitis, or rapid respiratory failure [5]. An example of a cancer-associated antigen in a clinical trial testing its efficacy in treating patients with solid tumors is human telomerase reverse transcriptase (hTERT). The function of hTERT is to support cancer cell growth and survival, predominantly through the maintenance of telomeres to promote cancer cell immortality [6]. In mouse models, vaccination against hTERT increased infiltration of T cells into B16 melanomas [7]. However, human trials still need to be performed.

Tumor-specific antigens, or neoantigens, are not found in non-cancerous cells. The immune system would recognize these antigens as nonself and "are less likely to induce autoimmunity compared to [tumor-associated antigens]" [5]. A promising new avenue for treatment for head and neck squamous cell carcinoma comes from research around antigens specific to human papilloma virus (HPV)-related cancers such as the oncoproteins E6 and E7 [5]. Oncoproteins have the ability to transform a cell into a tumor if they are introduced. Tumor cells are genetically unstable because of the repeated mutations of DNA that accumulate as cancer grows.

These mutations can be classified into two groups: driver mutations, which mainly contribute to cancer development; and passenger mutations, which are not involved with the disease's further activity or progression [5]. Numerous studies are now considering neoantigens from driver mutations shared among patients and others examining combinations of passenger mutations; however, it is reportedly difficult to predict these tumor-specific antigens. To address this issue, growing calls for the targeted sequencing of cancer-related gene mutations and building of an inventory of shared neoantigen peptide libraries of common solid tumors have emerged. These databases of neoantigens have reduced the time from prediction to patient vaccination, which has been determined crucial in progressing chronic disease.

The primary objective of this review was to characterize the advances in cancer-based therapeutics that considered DNA- and RNA-based nucleic acid therapies including vaccines, vector-based strategies, and genome injection. Secondarily, emerging evidence in cancer-based immunotherapies were also characterized. Additionally, these findings were contextualized when considering current clinical evidence for existing and alternate therapeutics.

## 2. Materials and Methods

To do this, we performed a narrative review utilizing databases including PubMed/MEDLINE, EMBASE, and Google Scholar with no time, setting, or language restrictions imposed on the search strategy. Primary research articles, including pre-clinical/animal trials, case studies, and non-primary studies, including systematic reviews and meta-analyses, were additionally included. We excluded secondary literature, in vitro, and experimental studies. Primary keywords utilized in our search included: "nucleic acid", "DNA", "RNA", "viral", "vaccine", "oncolytic", "checkpoint inhibitors", "immunotherapy", "therapeutic", "genome", "vector", "plasmid", and "cancer".

## 3. Results

### 3.1. Oncolytic Virus Therapy

Oncolytic virus therapy uses natural or genetically engineered viral vectors with redefined properties to only target and kill cancer cells as the active drug reagent, without damaging host tissue [3] (Figure 1). It is now recognized that in the majority of cancer cells, protection mechanisms against viral infection (including interferon-beta (IFN-β) signalling pathways) are damaged, which allows the virus to replicate and create damage in higher

concentrations [7]. Oncolytic viruses are becoming a popular anticancer therapy because of their high selectivity to infect and lyse tumor cells [8]. Findings from 1991 have suggested that in order to use viruses for cancer therapy in a controlled manner, the viral genome needs to be redesigned and reconstructed in order to control the vicious replication cycles that are naturally embedded in viral genomes [9].

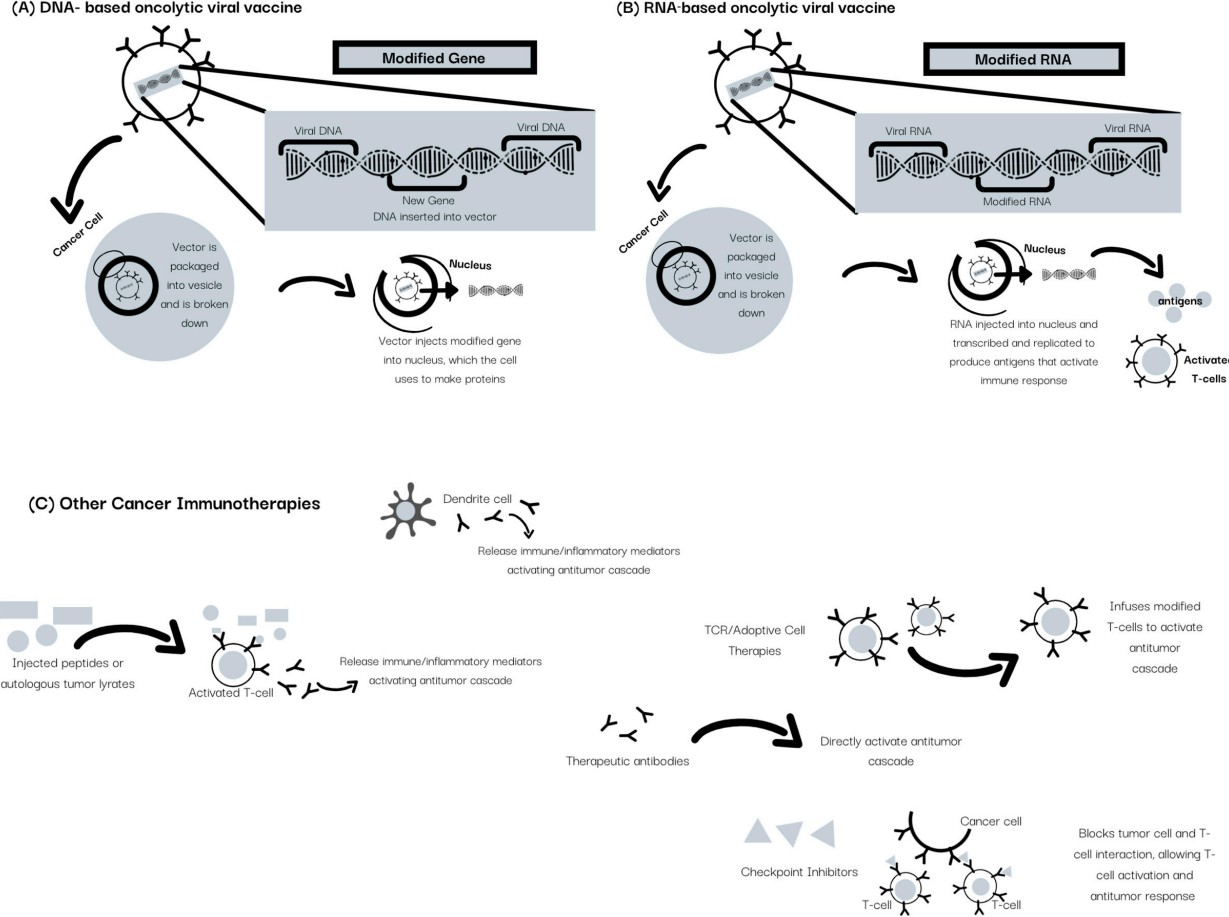

**Figure 1.** Oncolytic viral vaccines and selected cancer immunotherapies. (**A**) DNA-based oncolytic viral vaccines involve the insertion of a modified gene into the vaccine and its insertion into the nucleus to influence downstream signaling and production of proteins generating an anti-tumor response; (**B**) RNA-based oncolytic viral vaccines involve a similar mechanism involving the insertion of modified RNA into the viral RNA and its insertion into the nucleus to influence downstream transcription of antigens that generate the anti-tumor response; (**C**) other cancer immunotherapies include therapies utilizing injected peptides or autologous tumor lysates, dendritic cells, T-cell receptor (TCR)/adoptive cell therapies, therapeutic antibodies, checkpoint inhibitors, etc. (Designed by Reinette Schabert).

Oncolytic virus therapy uses natural or genetically engineered viral vectors with redefined properties to only target and kill cancer cells as the active drug reagent, without damaging host tissue [3] (Table 1). It is now recognized that in the majority of cancer cells, protection mechanisms against viral infection (including interferon-beta (IFN-β) signalling pathways) are damaged, which allows the virus to replicate and create damage in higher concentrations [7]. Oncolytic viruses are becoming a popular anticancer therapy because of their high selectivity to infect and lyse tumor cells [8]. Findings from 1991 have suggested that in order to use viruses for cancer therapy in a controlled manner, the viral genome needs to be redesigned and reconstructed in order to control the vicious replication cycles that are naturally embedded in viral genomes [9].

**Table 1.** Advantages and disadvantages of DNA-viral therapies, RNA-viral therapies, and immunotherapies for cancer treatment.

| Cancer Treatment Therapies | Advantages | Disadvantages |
|---|---|---|
| DNA-viral therapy | • Decreased risk of developing new tumours in patients with cancer relapse [10]<br>• Have easy-to-manipulate genomes [10]<br>• Thermodynamically favourable diffusion across as membranes, better for chemical delivery approaches [11]<br>• Larger half-life and heat stability compared to mRNA vaccines [11] | • Pre-existing immunity to viral backbones may hinder treatment efficacy [12]<br>• Hybrid structure of DNA vaccines requires improvement—fewer prokaryotic segments facilitate easier transfection into host cells [13] |
| RNA-viral therapy | • Can be synthesized rapidly and cost-effectively compared to DNA vaccines [11]<br>• Ability to encode more than one antigen within the genome sequence [11]<br>• Reduced risk of mutations due to the integration of the nucleic acid into the cell's genes and toxicity of built-up mRNA [14] | • Difficult to determine viral burden and toxicity for immunocompromised patients with cancer [11]<br>• Pre-existing immunity to viral backbones may hinder treatment efficacy [12] |
| Immunotherapies | • Widely adaptive in the management of multiple types of tumors [15]<br>• Improved survival rates long-term [16]<br>• Can initiate body's immune response to restore immune function and subsequently tumour cells [17] | • Defining a tumour immunogenicity and subsequent microenvironment conditions remain challenging [16]<br>• Utilization of immunocheckpoint inhibitors may cause negative regulation leading to autoimmune disease [17]<br>• Relatively inaccessible and high in treatment costs [17]<br>• Therapeutically novel and demonstrated efficacy among very small group of human participants [18] |

*3.2. Viruses with DNA Genomes*

Herpes simplex virus type 1 (HSV-1) has a linear double-stranded DNA genome embedded in an icosahedral capsid. These vectors affect viral replication, neuropathogenicity, and immune evasiveness, a strategy used by pathogenic organisms to increase the probability of being transmitted to the next available host [19,20] (Figure 1A). Attenuated HSV vectors are used to develop live viral vaccines, which deliver transgenes to the nervous system [21]. A major advantage to using HSV for viral therapy is that the entire genome has been sequenced using molecular biological techniques, which has led to the development of many potential therapeutic interventions for human health [21]. Additionally, HSV is a useful delivery mechanism for expressing human genes targeting cells of the nervous system due to its natural neurotropic feature, which means patients would only require very low doses of this virus to be inoculated into their cells [22]. HSV is also a non-integrative virus, which means that insertional mutagenesis is not an issue (this is not commonly seen with other viral vectors) [22]. Two examples of HSV-1 viral therapies include T-Vec, which is used to treat melanoma (skin cancer); and G47Δ, which is used to treat glioblastomas (cancer in the brain or spinal cord) [3].

Live vaccinia viruses (VACV) are enveloped viruses which contain a linear, double-stranded DNA genome. This virus belongs to the poxvirus family and is most well-known for being used by Edward Jenner in 1796 to cure smallpox, caused by the Variola virus [23]. VACV are desirable to be used in oncolytic viral therapies because of its high intravenous

stability for delivery, strong cytotoxicity and safety [24]. During the replication phase, VACV is able to accurately target and lyse tumor cells that spread through cancerous tissues [8]. An example is JX-594, which is used to treat advanced stage hepatocellular carcinomas, a type of liver cancer [3].

Adenoviruses have a double-stranded DNA genome, with a non-enveloped icosahedral-shaped nucleocapsid. They have many applications for gene therapy and cancer treatments. The immune responses activated by adenoviruses can be utilized cancer therapy in a number of ways. An advantage of using adenoviruses for treatment is the decreased risk of developing new tumors in patients with cancer relapse [10]. Adenoviruses have genomes that are easy to manipulate, which accept a large variety of transgene DNA insertions. It also has the benefit that these viruses do not replicate in host cells. However, it should be noted that because there is pre-existing immunity in humans, potential vaccine and gene therapy interventions should be aware of manipulating the viral vector in a way that becomes unrecognizable to the human host [12]. An example of an adenovirus vector is CG0070, which is a treatment used for non-muscle invasive bladder cancer [3].

### 3.3. Viruses with RNA Genomes

*Reovirus* is a stable, non-enveloped, double-stranded RNA virus (Figure 1B). It is considered as a benign human pathogen, as 50% of the population possesses antibodies to reovirus [25]. These viruses have been observed to have potent oncolytic properties that have been linked to various aberrations in cancer cells, including involvement in Ras oncogene signalling and impaired type I interferon (IFN) pathways [26]. Treatments using reovirus have been shown to promote the healthy secretion of a diverse range of proinflammatory cytokines and chemokines, which are capable of initiating strong antigen-specific T-cell responses [27]. Reolysin comes from the reovirus, which is used to treat metastatic or recurrent head and neck cancer [3].

*Newcastle disease virus (NDV)* is a fatal virus that primarily affects birds. NDV is negatively polarized and contains a non-segmented RNA genome [28]. NDV has the ability to selectively infect tumorigenic cells, while eliciting a strong immune response that can lyse the targeted cells without causing damage to the host's normal cells [29]. An immunotherapy using autologous tumor vaccine and NDV has been developed for patients with colorectal cancer. This therapy has been tested in clinical trials with positive and effective results [30].

*Polioviruses* are made of a protein capsid shell containing a positive-sense, single-stranded RNA genome. The use of oncolytic poliovirus has shown efficient tumor regression and selective cytotoxicity in animal models that were inoculated with attenuated poliovirus [31]. RNA viruses, such as poliovirus, are being used for anti-cancer treatment as they have shown effective viral propagation and invasion in vivo [32]. Polioviruses are able to selectively target cancerous cells in motor neurons by identifying the distribution of its cellular receptor, CD155, from the immunoglobulin (Ig) superfamily [33].

*Measles virus (MV)* is an enveloped, single-stranded RNA virus. MV is a common viral vector used for glioma therapy, as CD46 is a common biomarker that is overexpressed in these patients [11]. Although highly pathogenic, MV has natural oncolytic specificity that can be used for glioma therapy [34]. It has been shown that vaccines containing attenuated MV have been very successful in terms of safety. In very few cases of immunocompromised patients, measles vaccines have caused disease in the recipient, when the vaccine was administered as an oncolytic agent [35]. Attenuated MV has a wide array of receptor tropisms. CD46 is an important receptor used for gaining entry into the cell. This receptor serves as protection for autologous cells by blocking the C3 activation site during the complement cascade [36]. CD46 is also responsible for inducing cell-to-cell fusion between a virally infected cell and their neighbouring cells. When MV is used as a vaccine for a patient, this increases the concentration of CD46 receptors, which initiates the process of targeted cell death or apoptosis [37].

*3.4. Cancer Immunotherapy*

Cancer immunotherapy techniques are proving to be practical tools in the fight against the disease (Figure 1C). Checkpoint inhibitors' effectiveness in treating metastatic melanoma and adoptive T-cell therapy, with chimeric antigen receptor T cells, treats B-cell-derived leukemias and lymphomas, which are only two examples of advances that are reshaping clinical cancer treatment [15].

These changes result from extensive research into the complex and interconnected cellular and molecular processes that regulate immune responses over several years [15]. The discovery of cancer mutation-encoded neoantigens, advancements in vaccine production, advancement in cellular therapy delivery, and remarkable achievements in biotechnology are all promising developments [15].

Many clinical studies are currently underway to evaluate the possible synergistic effects of treatments that combine immunotherapy and other therapies [38]. New immune biomarkers and the ability to evaluate therapy responses by noninvasive testing can enhance cancer detection and prognosis in the early stages. Individualized immunotherapy focused on genetic, molecular, and immune profiling is a theoretically attainable target throughout the future.

The only approved cellular cancer therapy based on dendritic cells is sepulture-T. One method of inducing dendritic cells to present tumor antigens is by vaccination with autologous tumor lysates or short peptides [39]. These peptides are often given in combination with adjuvants to increase the immune and antitumor responses. Other adjuvants include GM-CSF and similar proteins. The most common source of antigens used for dendritic cell vaccine in glioblastoma as an aggressive brain tumor were whole tumor lysate, CMV antigen RNA, and tumor-associated peptides such as EGFRAII [39].

As clinical science advances, immunotherapy is becoming more widely available in clinical trials for early-stage cancers or as a first-line treatment choice [18]. On the other hand, many patients are unaware of recent immunotherapy breakthroughs and the increasing number of opportunities to engage in new cancer clinical trials. This is particularly true when we consider advances in therapeutic antibodies (e.g., monoclonal antibodies, antibody–drug conjugates, and bispecific antibodies), checkpoint inhibitors (e.g., PD-1/PD-L1 inhibitors), adoptive cell therapies (e.g., CAR-T), and TCR therapies.

Immunotherapy has the ability to produce long-term effects, but only a small percentage of patients respond currently. Primary and secondary immunotherapy resistance have a variety of causes, including tumor intrinsic factors and the dynamic interplay between cancer and its microenvironment [40]. Thus, the design of novel drugs and combination therapies requires (i) direct manipulation of the tumor or (ii) indirect immunogenic enhancement by altering the microenvironment. By systematically addressing the factors that mediate resistance, we can identify mechanistically driven novel approaches to improve immunotherapy outcomes [40].

In addition to surgery, chemotherapy, targeted pathway inhibition, and radiation therapy, immunotherapy has emerged as a standard pillar of cancer treatment. Immune checkpoint inhibitors (ICIs) such as those targeting cytotoxic T lymphocyte-associated protein 4 (CTLA-4) and programmed cell death protein 1/programmed cell death ligand 1 (PD-1/PD-L1) have been integrated into the standard of care regimens for patients with advanced melanoma, Merkel cell carcinoma, non-small cell lung cancer, cutaneous squamous cell carcinoma, urothelial cancer, renal cancer, refractory Hodgkin lymphoma, hepatocellular carcinoma, gastric cancer, triple-negative breast cancer, and microsatellite instability (MSI)-high tumors [41]. Beyond checkpoint inhibitors, cellular therapy in the form of chimeric antigen receptor (CAR) T cells directed at CD19 are now approved in patients with refractory B cell acute lymphoblastic leukemia and large B cell lymphoma [41]. Novel indications and integration of immunotherapy into earlier stages of the disease are being actively investigated.

While defining a tumor's immunogenicity and its microenvironment is challenging, clinical studies have validated several biomarkers. Tumor intrinsic factors, including

PD-L1 expression, 11 tumor mutation burden, and mismatch repair deficiency 14, are clinically useful yet imperfect biomarkers because they center around tumor cells [42]. We now recognize the pivotal role of the microenvironment, and emerging predictors of responsiveness to anti-PD-1 immunotherapy include associations with T cell receptor (TCR) diversity and/or clonality, host HLA genotype 16, a favorable gut microbiome, and even body mass index, possibly mediated by leptin, among other factors.

## 4. Discussion

Emerging evidence in oncolytic therapeutics have been categorized into three major vaccine-based platforms: peptide vaccines, nucleic acid vaccines, and cell-based vaccines. While peptide vaccines are popular, only a few clinical trials have been completed, and future steps are still being considered. Short peptides do not require processing by APCs, offering an advantage to longer peptide chains which must be taken up and processed by antigen-presenting cells. Although this ability may seem appealing, clinical trials have suggested an increased risk of dysfunction in the immune system if short peptide chains stimulate T cells without the presence of other stimulating factors or adjuvants. APCs with long peptide chain-derived antigens can thus activate CTL or helper T cells without inducing energy by transmitting signals via both T cell receptors (TCRs) and co-stimulatory molecules, avoiding the problem altogether [18].

Cell-based vaccines involve the use of the individual patient's cancer cells. Irradiated cells from the tumor of the cancer patient are administered via vaccine along with an adjuvant, similar to how peptide-based vaccines work. The adjuvant is able to provide an inflammatory context for antigen presentation, so that the T cells are stimulated while being provided with the irradiated tumor cell, which potentially has many antigens that could be targeted [43]. This approach has been tested with many different types of cancers: colorectal cancer, lung cancer, renal cell carcinoma, prostate cancer, and melanoma [18]. Moreover, these cells may be genetically modified such that there is an increased capacity to produce cytokines, signalling proteins to facilitate inflammation, as well as a granulocyte-macrophage colony-stimulating factor (GM-CSF), which may also act as cytokines that stimulate the production of granulocytes and monocytes. GVAX is a vaccine for cancer that uses genetically modified cancer cells that secrete GM-CSF after a patient goes through radiation treatment. The purpose of the vaccine is to stop the uncontrollable growth of the tumor, thereby extending the life of the individual. Early clinical trials in phases one and two have found good results in non-small-cell lung carcinoma; however, no effects have been seen in phase three clinical trials for prostate cancer [18].

Another cell-based avenue that is being explored is the possibility of using dendritic cells (DCs) as the vehicle for a variety of antigens, including tumor cells, tumor-derived proteins or peptides, and DNA/RNA [18]. DCs are most often found in places that are likely to see exposure to the environment, including lungs, digestive tract, and the skin [43]. When cancer cells die occasionally as a result of nutrient deprivation, they get engulfed by DCs or other APCs. These cells are not only able to initiate the phagocytosis of the dying cancer cell and stimulate T cell activation, but also possess the capacity to sense abnormalities in their environment due to cell surface receptors that recognize the signals released by an apoptotic or necrotic cell. The signals as a whole are referred to as damage-associated molecular patterns (DAMPs). DCs also have an important role to play in immunotherapeutics because they are able to absorb and express tumor-associated antigens [38]. DC-based mRNA vaccine therapies now account for a vast majority of mRNA cancer vaccines in clinical trials. For example, the Stipuleucel-T vaccine containing prostatic acid phosphatase (PAP) and GM-CSF was developed utilizing autologous cells from the body of the patient, and as PAP is specific to prostate cancer, Stipuleucel-T represents an emerging immunotherapy for castration-resistant prostate cancer [15].

Nucleic acid-based vaccines have several advantages over other vaccine platforms because they will enable the delivery of multiple antigens covering various somatic tumor mutations, eliciting both humoral and cell-mediated immune response [14]. The nucleic

acids can hold the genetic code to produce multiple antigens in one vaccine. While peptide vaccines are time and labor-intensive to produce, nucleic acid vaccines are inexpensive and can be synthesized stably [18].

Another major advantage that nucleic acid-based vaccines have when compared to peptide vaccines is the effectiveness of viral vectors that can be embedded into a genetic code. The nucleic acids are thus taken into the cells much more efficiently, as the immune system does not identify the viral vector as a foreign pathogen. However, Igarashi and Sasada (2020) warn that it may be difficult to repeatedly administer viral vectors due to the induction of antiviral immune responses.

While DNA and RNA-based viral therapeutics each has advantages and disadvantages, RNA-based methods may hold greater promise for researchers because, unlike DNA, they do not need to penetrate the nuclear membrane and can function when delivered to the cytoplasm of APCs [5]. Messenger RNA (mRNA) platforms have been experiencing a considerable burst in preclinical and clinical research with over twenty mRNA-based immunotherapies having been entered in clinical trials with some promising outcomes in solid tumor treatments [14]. There have been exciting innovations with mRNA-based vaccines that are not related to cancer, namely the FDA approval of Pfizer-BioNTech and Moderna COVID-19 vaccines.

This rise in interest in mRNA vaccine technology will undoubtedly lead to innovations and research for both infectious diseases and cancer mRNA-based vaccines. A specific type of mRNA is being explored that will maximize the vaccine's effect in terms of the length of time and magnitude of the production of antigens. Self-amplifying mRNA (SAM) originates from viruses that carry single-stranded mRNA. It consists of two main regions or open reading frames, one that encodes the antigen sequence and the other that encodes proteins and structures that will amplify the mRNA and the immune response to the growing number of antigens. The strand has been modified so that the genes which encode the viral particles and structures are replaced with genes encoding the antigen of interest [14]. While the genetic instructions for the viral infectious parts are removed from the RNA strand, the instructions for the replication machinery remain and enable the RNA to be amplified within the cytoplasm of a cell.

Unfortunately, the challenge to using a viral vector lies in finding the tolerable viral burden and toxicities for people with cancer who already have a compromised immune system, and the potential for pre-existing immunity to many of the viral backbones that are currently deemed safe in humans, which will ultimately affect the efficacy of the therapy [43]. It has also been suggested that a 64-fold lesser dose of SAM achieved the equivalent immunity to the non-replicating mRNA [14]. Furthermore, the superiority of mRNA-based vaccines has been cited as three-fold: (i) the ability to encode more than a particular antigen; (ii) the ease of integration into the cell and the high degradability of mRNA by RNAs compared to DNA and the reduced risk of mutations due to the integration of the nucleic acid into the cell's genes or the toxicity of built-up mRNA; and (iii) its ability to be synthesized rapidly and in a cost-efficient, scalable manner.

The continuous development of safer and more effective oncolytic viral therapies is expected to play a critical role in prolonging the survival of cancer patients [3]. However, the manufacturing process involved in creating oncolytic viruses tends to be very difficult, as it is associated with the effectiveness of the viral delivery mechanism and, ultimately, the success of the oncolytic virotherapy treatment [44]. There are many individual barriers associated with the immunosuppressive tumors, which can be easily missed by immune surveillance. This is a critical limitation in cancer therapy, as strong immune surveillance is required to prevent the rapid proliferation and dissemination of cancerous cells to other organs and tissues [44].

Further research is needed to determine whether current oncolytic viral therapies are affected, in terms of effectiveness, by the presence of pre-existing immunity to viruses. It has been suggested that some viral therapies, such as NDV, are not as effective in targeting tumors because the virus elicits too strong of an immunological response when

administered into the patient [45,46]. Immunosuppression is a plausible solution, as it can help localize tumor-targeting specificity, thereby increasing the effectiveness of oncolytic viral therapies.

## 5. Conclusions

Oncolytic viral therapies and immunotherapies are both emerging and promising therapeutic avenues in the treatment of cancers. These therapies can be classified by vector delivery systems or based on their mechanisms of action. DNA-based and RNA-based viral therapies have unique structural characteristics to enable the propagation and vaccination of individuals. Nonetheless, host factors and immunity are limitations to both of these modalities. In contrast, immunotherapies have a wide array of utility and function via a number of adaptive mechanisms. They have significantly enhanced survival statistics and been demonstrated to reverse several toxic effects of tumors. However, it is key to note that checkpoint inhibitors and other novel therapy are limited by negative regulation, immunogenicity, accessibility, cost, and limited research. As such, future research warrants a focus on both targeted and adjuvant therapies to increase efficacy of current therapeutics.

**Author Contributions:** Conceptualization, P.A.J., A.W., J.C.J., T.W., and A.M.; methodology, P.A.J. and A.W.; software, P.A.J.; validation, P.A.J., A.W., T.W., and J.C.J.; formal analysis, P.A.J., A.W., T.W., and J.C.J.; investigation, P.A.J., A.W., T.W., J.C.J and F.F.; resources, P.A.J. and A.M.; data curation, P.A.J. and A.W.; writing—original draft preparation, P.A.J. and A.W.; writing—review and editing, P.A.J., J.C.J., Z.S., and F.F.; visualization, P.A.J., R.S. and F.F.; supervision, P.A.J.; project administration, P.A.J., J.C.J., and A.M. All authors have read and agreed to the published version of the manuscript.

**Funding:** This research received no external funding.

**Institutional Review Board Statement:** Not applicable.

**Informed Consent Statement:** Not applicable.

**Data Availability Statement:** Not applicable.

**Acknowledgments:** We would like to acknowledge the technical support received from the Government of Canada and the Antarctic Institute of Canada.

**Conflicts of Interest:** The authors declare no conflict of interest.

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
