# Peer review of "Advances in DNA- and RNA-Based Oncolytic Viral Therapeutics and Immunotherapies"

_2673-8007, doi:10.3390/applmicrobiol2020024_

Round 1
Reviewer 1 Report
Cancer continuously to be a worldwide problem that affects human health and/or lifestyle. Among the advances in different types of therapeutics to treat cancer, viral-based therapeutics provide an option to control and/or treat certain types of cancer.
The following review provides a basic summary of a subset of DNA- and RNA-based oncolytic viral platforms being used. The review covers the main points of the authors' goals for the review, however, some basic additions to this review will provide a stronger manuscript:
*Different types of Learners utilize visual aids to help learn a particular topic. Authors need to implement 2-4 figures/graphics and/or tables to help organize/visualize the written material covered. This will help emphasize key take-home points the authors want readers to understand from this review and from some of the written material provided here. Also, review papers usually include 1-4 figures/graphics/tables at a minimum to help organize the written material, so this is quite common.
-Biorender is a web-based platform that can be used for free figure development, as an example of image-based software that can be easily used.
*Grammar review of the abstract and other written sections is suggested, as there are some areas that need minor refinement.
*Section 2. Materials and Methods section- This section is not usually included in review-based papers. Please look at other recent, previously-published MDPI- and/or Appl. Microbiol.-based review papers and identify if this section is needed for review-based papers.
Author Response
Cancer continuously to be a worldwide problem that affects human health and/or lifestyle. Among the advances in different types of therapeutics to treat cancer, viral-based therapeutics provide an option to control and/or treat certain types of cancer.
The following review provides a basic summary of a subset of DNA- and RNA-based oncolytic viral platforms being used. The review covers the main points of the authors' goals for the review, however, some basic additions to this review will provide a stronger manuscript:
*Different types of Learners utilize visual aids to help learn a particular topic. Authors need to implement 2-4 figures/graphics and/or tables to help organize/visualize the written material covered. This will help emphasize key take-home points the authors want readers to understand from this review and from some of the written material provided here. Also, review papers usually include 1-4 figures/graphics/tables at a minimum to help organize the written material, so this is quite common.
-Biorender is a web-based platform that can be used for free figure development, as an example of image-based software that can be easily used.
- Response: Thank you, we have added a figure and table.
*Grammar review of the abstract and other written sections is suggested, as there are some areas that need minor refinement.
- Response: Thank you, we have used a proofreading service.
*Section 2. Materials and Methods section- This section is not usually included in review-based papers. Please look at other recent, previously-published MDPI- and/or Appl. Microbiol.-based review papers and identify if this section is needed for review-based papers.
- Response: Thank you, this section has been removed.
Reviewer 2 Report
The article focused on “Advances in DNA- and RNA-based oncolytic viral therapeutics”. The topic is very interesting and of clinical significance. There are some suggestions for this article.
- In abstract line 13-15 “The prevalence of cancer diagnoses is increasing globally, and treatments are needed to support the increasing body of patient care.” The increasing is disease but not diagnosis. Treatment might not be used to support patient care. Suggest to revise this sentence.
- The same for line 32-33 in the Introduction part: “symptoms associated with potentially fatal diagnoses.” What cause fatality is disease but not symptoms or diagnosis. Suggest to revise this sentence.
- In the Introduction part, line 36-37 “Other options for cancer treatment are becoming available for patients that provide much safer, but still effective, interventions.” Please clarify this sentence and cite the reference.
- Line 52-53 “is telomerase (hTERT).” Please show the full name of this term when first mentioned.
- Line 55-56 “In mouse models, vaccination against hTERT increased infiltration of T cells into B16 melanomas though human trials still need to be performed.” Please clarify this sentence and cite the reference.
- Line 57 and 62 “Tumour” should be Tumor
- Line 59-62 “A promising new avenue for treatment for head and neck squamous cell carcinoma…oncoproteins E6 and 61 E7…. if they are introduced.” Please cite the reference.
- Line 80 and 89: Since it is a review article but not original or meta-analysis article, it is not suitable for “Materials and Methods” and “Result”. Instead the sub-heading of each paragraph is suggested.
- Line 122-123 “HSV is a useful delivery mechanism for expressing human genes targeted towards the cells of the nervous system” Please describe it more detail and cite the reference.
- Line 133 “VACV are desirable to use in oncolytic viral therapies” would be best revised as “VACV are desirable to be used in oncolytic viral therapies”
- Line 141-142 “The strong immune responses from adenoviruses can be stimulated for use in the field of cancer therapy.” Please describe it more detail and cite the reference.
- For 3.4. Cancer Immunotherapy line 191-260. I am not sure what is the correction between this paragraph with the title “Advances in DNA- and RNA-based oncolytic viral therapeutics”?
- For 4. Discussion. Line 261-367. Suggest to express the viewpoint regarding this topic but not just summarize previous paragraph.
- Suggest to add a “Graphic Abstract”
- Suggest to make a table illustrating the advantages and disadvantages of these oncolytic viral therapeutics.
Author Response
- In abstract line 13-15 “The prevalence of cancer diagnoses is increasing globally, and treatments are needed to support the increasing body of patient care.” The increasing is disease but not diagnosis. Treatment might not be used to support patient care. Suggest to revise this sentence. Thank you, this has been revised.
- The same for line 32-33 in the Introduction part: “symptoms associated with potentially fatal diagnoses.” What cause fatality is disease but not symptoms or diagnosis. Suggest to revise this sentence. Thank you, this has been revised.
- In the Introduction part, line 36-37 “Other options for cancer treatment are becoming available for patients that provide much safer, but still effective, interventions.” Please clarify this sentence and cite the reference. Thank you, this has been revised.
- Line 52-53 “is telomerase (hTERT).” Please show the full name of this term when first mentioned. Thank you, this has been revised.
- Line 55-56 “In mouse models, vaccination against hTERT increased infiltration of T cells into B16 melanomas though human trials still need to be performed.” Please clarify this sentence and cite the reference. Thank you, this has been revised.
- Line 57 and 62 “Tumour” should be Tumor Thank you, this has been revised.
- Line 59-62 “A promising new avenue for treatment for head and neck squamous cell carcinoma…oncoproteins E6 and 61 E7…. if they are introduced.” Please cite the reference. Thank you, this has been revised.
- Line 80 and 89: Since it is a review article but not original or meta-analysis article, it is not suitable for “Materials and Methods” and “Result”. Instead the sub-heading of each paragraph is suggested. Thank you, this has been revised.
- Line 122-123 “HSV is a useful delivery mechanism for expressing human genes targeted towards the cells of the nervous system” Please describe it more detail and cite the reference. Thank you, this has been revised.
- Line 133 “VACV are desirable to use in oncolytic viral therapies” would be best revised as “VACV are desirable to be used in oncolytic viral therapies” Thank you, this has been revised.
- Line 141-142 “The strong immune responses from adenoviruses can be stimulated for use in the field of cancer therapy.” Please describe it more detail and cite the reference. Thank you, this has been revised.
- For 3.4. Cancer Immunotherapy line 191-260. I am not sure what is the correction between this paragraph with the title “Advances in DNA- and RNA-based oncolytic viral therapeutics”? Thank you, this has been revised.
- For 4. Discussion. Line 261-367. Suggest to express the viewpoint regarding this topic but not just summarize previous paragraph. Thank you, this has been revised.
- Suggest to add a “Graphic Abstract” Thank you, we have added a figure instead.
- Suggest to make a table illustrating the advantages and disadvantages of these oncolytic viral therapeutics. Thank you, this has been revised.
Round 2
Reviewer 1 Report
The revised manuscript is much more inviting with the included table and figure. Also, extra content has been added to help clarify subject areas throughout the text. However, there a few more items that would make the manuscript even stronger to readers:
*Line 70: HPV needs to be defined before abbreviated. This has been followed fairly consistently in the text, however, please double check for more of these in the text.
*Figure 1- Double check the wording in the figure. For example, “RNA based” should be “RNA-based” like the already listed “DNA-based”. Also, increasing font size of this figure will be tremendously helpful for readers.
*Table 1- Be sure to reference the information in the table, as this is a Review article and not a Perspective article. It may be referenced in the main text already, but each point should be referenced within the table or in the title of the table so readers can pinpoint where the information came from quickly.
*The viruses listed in 3.3 and 3.4 are listed differently and need to be uniformly listed. For example, some are bold and some are italicized.
*Be sure all text content is the same font/spacing/size. In Section 4 and the Discussion, there is written content that does not match most of the text content in the manuscript.
*Line 213- What does "n.d." stand for here? Is this a proper citation?
Author Response
The revised manuscript is much more inviting with the included table and figure. Also, extra content has been added to help clarify subject areas throughout the text. However, there a few more items that would make the manuscript even stronger to readers:
*Line 70: HPV needs to be defined before abbreviated. This has been followed fairly consistently in the text, however, please double check for more of these in the text.
Thank you, this has been modified.
*Figure 1- Double check the wording in the figure. For example, “RNA based” should be “RNA-based” like the already listed “DNA-based”. Also, increasing font size of this figure will be tremendously helpful for readers.
Thank you, this has been modified.
*Table 1- Be sure to reference the information in the table, as this is a Review article and not a Perspective article. It may be referenced in the main text already, but each point should be referenced within the table or in the title of the table so readers can pinpoint where the information came from quickly.
Thank you, this has been modified.
*The viruses listed in 3.3 and 3.4 are listed differently and need to be uniformly listed. For example, some are bold and some are italicized.
Thank you for this suggestion. The main goal of 3.3 is to summarize key examples of viral-based vaccines, whereas the objective of 3.4 is to provide an overview of immunotherapies, hence the viruses are not the focus.
*Be sure all text content is the same font/spacing/size. In Section 4 and the Discussion, there is written content that does not match most of the text content in the manuscript.
Thank you, this has been modified.
*Line 213- What does "n.d." stand for here? Is this a proper citation?
Thank you, this has been modified.
Reviewer 2 Report
had been revised as suggestions
Author Response
had been revised as suggestions
-Thank you for your comments.